# Decoding oncogenic secrets of regulator of chromosome condensation 1: A breakthrough mechanistic evidence from breast and lung cancer models

**Mohamed El-Tanani[1]\*, Shakta Mani Satyam[2], Syed Arman Rabbani[3], Razan M. Obeidat[4], Yahia El-Tanani[5], Alaa A. A. Aljabali[6], Taher Hatahet[7]\***

**1** RAK College of Pharmacy, RAK Medical and Health Sciences University, Ras Al Khaimah, United Arab Emirates, **2** Department of Pharmacology, RAK College of Medical Sciences, RAK Medical and Health Sciences University, Ras Al Khaimah, United Arab Emirates, **3** Department of Clinical Pharmacy, RAK College of Pharmacy, Ras Al Khaimah Medical and Health Sciences University, Ras Al Khaimah, United Arab Emirates, **4** Pharmacological and Diagnostic Research Center, Faculty of Pharmacy, Al–Ahliyya Amman University, Amman, Jordan, **5** Royal Cornwall Hospital Trust, NHS, London, United Kingdom, **6** Department of Pharmaceutics and Pharmaceutical Technology, Faculty of Pharmacy, Yarmouk University, Irbid, Jordan, **7** Department of Pharmaceutical Technology, Queen's University Belfast, Belfast, United Kingdom

\* t.hatahet@qub.ac.uk (TH); eltanani@rakmhsu.ac.ae (ME)

## Abstract

The Regulator of Chromosome Condensation 1 (RCC1), a master regulator of cell cycle progression, chromatin structure, and nuclear transport, emerges as a powerful driver of cancer progression. Elevated RCC1 expression in breast and lung cancers is closely tied to enhanced tumor cell survival, proliferation, and metastasis, positioning it as a promising therapeutic target. This study unveils RCC1's pivotal role in cancer biology by silencing its expression in MDA-MB-231 (breast cancer) and A549 (lung cancer) cell lines using shRNA. RCC1 knockdown dramatically reduced cell viability, colony formation, and motility, while inducing apoptosis, as evidenced by increased apoptotic markers and reduced anti-apoptotic Bcl2 expression. Gene expression analysis revealed downregulation of cell cycle and DNA repair pathways, highlighting RCC1's critical role in sustaining oncogenic mechanisms. These findings underscore RCC1 as a gatekeeper of tumor survival, capable of resisting apoptosis and promoting metastasis. Targeting RCC1 offers a dual advantage: disrupting cancer growth and enhancing apoptotic pathways, creating an exciting opportunity for precision therapies. By illuminating RCC1's integration into survival networks, this study not only advances our understanding of cancer biology but also lays the groundwork for innovative treatments aimed at halting cancer progression and metastasis.

## Introduction

Cancer continues to be a leading cause of death worldwide, with breast and lung cancers being among the most prevalent cancers [1,2]. Breast cancer, particularly metastatic forms, remains a major cause of mortality in women worldwide [3]. Likewise, lung cancer, notably non-small

**Data availability statement:** All relevant data are within the manuscript.

**Funding:** The author(s) received no specific funding for this work.

**Competing interests:** The authors have declared that no competing interests exist.

cell lung cancer (NSCLC), poses a significant challenge owing to its aggressive behavior and late-stage diagnosis [4]. The high metastatic potential of these cancers complicates treatment and significantly affects the patient prognosis.

RCC1 (Regulator of Chromosome Condensation 1) is a well-conserved guanine nucleotide exchange factor (GEF) for the Ran protein, which is essential for various cellular processes such as nuclear transport, spindle assembly, and chromatin condensation. RCC1 catalyzes the conversion of RanGDP to RanGTP, establishing a RanGTP gradient essential for nuclear-cytoplasmic transport [5–11]. In non-cancerous cells, RCC1 plays a fundamental role in ensuring proper mitotic progression, maintaining genomic stability, and preventing aneu-ploidy, all of which are critical for normal cellular homeostasis [12–14]. However, its dysreg-ulation in cancer cells disrupts these tightly controlled processes, leading to unchecked cell proliferation and enhanced survival. RCC1's interactions with survival-promoting signaling cascades further support cancer cell resilience, emphasizing its potential role in oncogenesis.

Recent studies have identified RCC1 as a potential oncogene, with elevated expression lev-els observed in multiple cancer types, including breast, lung, ovarian, and colorectal cancers [15,16]. High RCC1 expression correlates with aggressive phenotypes and poorer clinical out-comes, suggesting its role in promoting cell proliferation, survival, and metastasis. Studies in breast and lung cancer models have linked RCC1 overexpression to increased tumor invasive-ness and resistance to apoptosis, indicating that RCC1 may be crucial for cancer cell survival under therapeutic stress [17,18].

RCC1's primary function in cell cycle regulation is particularly significant for cancer progression. By modulating the RanGTP gradient, RCC1 facilitates key transitions in the cell cycle, particularly in G1/S and G2/M phases. In cancer cells, overexpression of RCC1 disrupts these checkpoints, allowing uncontrolled progression through mitosis and contributing to rapid proliferation, a hallmark of tumorigenesis [19–21]. Furthermore, RCC1's ability to influ-ence apoptosis resistance is critical to its oncogenic role. Cancer cells often evade apoptosis by altering pro-survival signaling pathways, enabling them to survive under adverse condi-tions. RCC1 overexpression has been shown to enhance survival signaling by upregulating anti-apoptotic proteins such as Bcl2, which inhibits mitochondrial membrane permeabiliza-tion and prevents apoptotic pathway activation [22,23].

Despite growing evidence of RCC1's role in oncogenesis, its specific contributions to breast and lung cancer remain poorly understood. Given its involvement in central cancer progres-sion mechanisms such as cell cycle regulation, apoptosis resistance, and metastatic behavior, it is hypothesized that RCC1 may drive the aggressive phenotypes observed in these cancers [24,25]. By elucidating RCC1's role in breast and lung cancer, this study aimed to identify potential therapeutic pathways to target its oncogenic functions.

This study aimed to clarify RCC1's role in cancer progression through an in-depth analysis of its effects on cell survival, apoptosis, and metastasis in breast and lung cancer models. We hypothesized that RCC1 knockdown would lead to decreased cancer cell viability, migra-tion, and invasion, along with increased apoptotic responses. Specific objectives include (1) examining RCC1's impact on cancer cell proliferation and apoptosis, (2) evaluating its role in metastatic behaviors, and (3) investigating associated molecular pathways through gene expression profiling. The insights gained from this study may reveal novel therapeutic avenues for targeting RCC1 in cancer treatment.

## Materials and methods

### Cell lines and culture conditions

MDA-MB-231 breast cancer and A549 lung cancer cell lines were obtained from the American Type Culture Collection (ATCC) and used to investigate RCC1's role in cancer progression.

Cells were cultured in Dulbecco's modified Eagle's medium (DMEM) supplemented with 10% fetal bovine serum (FBS), 1% penicillin-streptomycin, and 2 mM L-glutamine and maintained at 37°C in a humidified atmosphere with 5% $CO_2$. Each experiment was conducted using cells within 20 passages to ensure consistency and reliability of results [5,6].

## Ethical approval and informed consent statement

This study did not require ethical approval as it was conducted using commercially available cell lines obtained from the American Type Culture Collection (ATCC). These cell lines are pre-established and were not derived from human or animal subjects in-house. Therefore, no ethical concerns related to human or animal experimentation were applicable.

Additionally, informed consent was not required, as the study did not involve human participants, tissues, or identifiable personal data. All procedures were performed in accordance with institutional and international guidelines for in vitro research.

## Transfection for RCC1 knockdown

To assess the role of RCC1 in cancer cell dynamics, RCC1 was knocked down using specific shRNA (small hairpin RNA) constructs. Two distinct shRNA sequences targeting RCC1 (shRCC1-A and shRCC1-B) and a scrambled shRNA sequence (shSCr) as a control were designed and synthesized. Transfections were performed using GeneJuice® transfection reagent following the manufacturer's instructions, and the cells were incubated for 72 h to allow adequate knockdown before analysis. Knockdown efficiency was confirmed using quantitative real-time PCR (qRT-PCR) and western blotting. To minimize potential off-target effects, the shRNA sequences were computationally screened for sequence specificity against the human transcriptome using BLAST analysis. Additionally, off-target effects were controlled by using two independent RCC1-targeting shRNAs and confirming knockdown specificity through quantitative real-time PCR (qRT-PCR) and western blotting.

## Quantitative real-time PCR (qRT-PCR)

Total RNA was extracted from transfected cells using the RNeasy Mini Kit (Qiagen), according to the manufacturer's protocol. cDNA was synthesized using an iScript cDNA Synthesis Kit (Bio-Rad) and used as a template for qRT-PCR. Primers for RCC1, Bcl2, and p53, as well as for the housekeeping genes GAPDH and β-actin, were designed for target specificity. qRT-PCR reactions were performed on a Bio-Rad CFX96 Touch Real-Time PCR Detection System using SYBR Green PCR Master Mix. The relative expression of each gene was calculated using the $2^{-\Delta\Delta Ct}$ method, normalized to GAPDH and β-actin as internal controls [26]. To control for off-target effects, non-targeting shRNA controls were included in every experiment, and gene expression changes were validated using multiple reference genes.

## Western blot analysis

To validate the RCC1 knockdown and examine the downstream effects on apoptosis-related proteins, we performed western blotting. Cells were lysed in radioimmunoprecipitation assay (RIPA) buffer supplemented with protease and phosphatase inhibitors. The protein concentration was determined using the Bradford assay, and 30 μg of protein per sample was separated on 10% SDS-PAGE gels and then transferred onto PVDF membranes. Membranes were blocked with 5% non-fat milk in TBS-T and probed with primary antibodies for RCC1, Bcl2, p53, and GAPDH (loading control) at 4°C overnight. After incubation with HRP-conjugated secondary antibodies, the proteins were visualized using enhanced chemiluminescence (ECL) and quantified using ImageJ software [27]. To control for potential off-target effects of

shRNA, we compared protein expression patterns between the two independent shRNAs and the scrambled control, ensuring consistency across knockdown conditions. Each assay was performed under uniform control conditions to allow comparisons among various experimental conditions, including scrambled shRNA sequences for negative controls.

## MTT assay for cell viability

The effect of RCC1 knockdown on cell viability was evaluated using the MTT assay. Transfected cells were seeded in 96-well plates at a density of $1.2–1.8 \times 10^4$ cells per well and allowed to grow for 24, 48, and 72 hours. At each time point, 20 μL of MTT reagent (5 mg/mL) was added to each well and the cells were incubated for 3 h at 37°C. The medium was carefully removed and 100 μL of DMSO was added to dissolve the formazan crystals. The absorbance was measured at 570 nm, with a reference wavelength of 690 nm. Cell viability was calculated relative to that of the control cells (without the knockdown of RCC1), with each experiment performed in triplicate [28].

## Colony formation assay

To assess the effect of RCC1 on long-term proliferation, a colony formation assay was performed. Following transfection, cells were seeded in 6-well plates at a low density (500 cells/well) and cultured for 10 days to allow colonies to form. The colonies were fixed with 4% paraformaldehyde, stained with 0.5% crystal violet, and counted under a microscope. Colony-forming efficiency was calculated as the percentage of seeded cells that formed colonies, comparing RCC1-knockdown groups to scrambled controls [27].

## Migration and invasion assays

To evaluate RCC1's effect on metastatic behavior, migration and invasion assays were conducted using Boyden chambers with an 8 μm pore polycarbonate membrane. For migration assays, transfected cells were suspended in serum-free DMEM and seeded into the upper chambers, whereas the lower chambers were filled with DMEM containing 10% FBS as a chemo-attractant. For invasion assays, the membranes were pre-coated with Matrigel to simulate the extracellular matrix. After 24 h, the cells that migrated or invaded through the membrane were fixed, stained with crystal violet, and counted in five random fields under a microscope [25].

## Flow cytometry for apoptosis analysis

Apoptotic cells were detected using flow cytometry with propidium iodide (PI) staining. Following RCC1 knockdown, cells were harvested, fixed in ice-cold 70% ethanol overnight, and stained with 50 μg/mL PI in the presence of RNase A. Samples were analyzed using a BD LSRII Flow Cytometer with data acquisition in the FL2 channel. The percentage of apoptotic cells was quantified by measuring the sub-G1 population, indicative of apoptotic DNA fragmentation, and was compared between RCC1-knockdown and scrambled cells [6].

## Gene expression profiling by microarray analysis

To investigate the transcriptional changes induced by RCC1 knockdown, a microarray analysis was performed. Total RNA was extracted from the transfected cells and processed for microarray hybridization on an Affymetrix GeneChip platform. Differential expression analysis was conducted to identify significantly altered genes, focusing on pathways related to cell cycle regulation, apoptosis, and metastasis. Normalization of raw

expression data was performed using the robust multi-array average (RMA) method to reduce technical variation. Differentially expressed genes were identified using a false discovery rate (FDR) threshold of < 0.05 and fold-change > 2. Functional enrichment analysis was conducted using DAVID Bioinformatics Resources to interpret pathway alterations. Selected genes showing significant changes were further validated by qRT-PCR to confirm these findings [29].

## Kaplan-Meier survival analysis of RCC1 expression in cancer

Kaplan-Meier survival analysis was utilized to assess the prognostic impact of RCC1 and its associated genes in cancer patients, leveraging data from the Kaplan-Meier Plotter database. This database, which aggregates survival data across extensive breast and lung cancer cohorts, enabled the stratification of patient samples into high and low expression groups for RCC1, SGOL2, and USP53 based on median expression values. Survival curves were then generated and analyzed for each gene using the log-rank test to assess statistical differences between groups. Hazard ratios (HR) with 95% confidence intervals (CI) were calculated to determine the relative risk associated with high versus low RCC1 expression. Additionally, multivariate Cox proportional hazards models were employed to adjust for confounding factors such as age, tumor grade, and treatment regimen, providing a more comprehensive evaluation of RCC1's prognostic significance. This approach allowed for a detailed evaluation of correlations between gene expression levels and survival outcomes, thereby offering insights into the prognostic value of RCC1 and its downstream targets.

## Statistical analysis

All statistical analyses were conducted using IBM SPSS Statistics version 29, with significance level set at $p < 0.05$. Data were expressed as mean ± SD for continuous variables. For gene and protein expression analyses (qRT-PCR and Western blot), independent sample t-tests were used to compare RCC1, Bcl2, and p53 levels between RCC1 knockdown (shRCC1-A and shRCC1-B) and scrambled control groups, normalized to GAPDH and β-actin. To account for potential batch effects and technical variability, a mixed-effects model was applied to validate reproducibility across independent experiments. To further validate the absence of off-target effects, gene expression changes were compared between two independent RCC1-targeting shRNAs to ensure reproducibility. In the MTT assay, a two-way ANOVA with Bonferroni post hoc tests examined RCC1 knockdown effects over time (24, 48, and 72 hours) on cell viability in MDA-MB-231 and A549 cells. Colony formation differences between knockdown and control groups were assessed using a one-way ANOVA with Tukey post hoc test. Migration and invasion assays were analyzed with independent sample t-tests on cell counts from five fields of view to determine significant differences between groups. Apoptotic cell percentages from flow cytometry (sub-G1 phase) were compared using independent sample t-tests. Principal component analysis (PCA) was performed to assess variance between experimental groups and confirm clustering of samples based on RCC1 expression levels. For microarray analysis, hierarchical clustering was used to visualize differential gene expression, and statistical significance was determined using the limma package in R, with an adjusted p-value threshold of < 0.05. Kaplan-Meier survival analysis, with log-rank tests on publicly available datasets, evaluated RCC1, SGOL2, and USP53 as prognostic markers in breast and lung cancer patient cohorts. Multivariate Cox regression models were used to assess the independent prognostic value of RCC1 while adjusting for clinicopathological covariates, such as tumor stage and lymph node involvement, thereby strengthening the reliability of survival predictions.

## Results

### Efficacy of RCC1 silencing in cancer cell lines

To evaluate RCC1's role in cancer cell survival and metastasis, its expression was selectively silenced in two cancer models: MDA-MB-231 breast cancer cells and A549 lung cancer cells. Using two distinct RCC1-targeting short hairpin RNAs (shRNAs), shRCC1-A and shRCC1-B, along with a scrambled shRNA control (shScr), we ensured specificity and minimized off-target effects. Post-transduction, six stable cell lines were established: MDA-MB231-shScr, MDA-MB231-shRCC1-A, MDA-MB231-shRCC1-B, A549-shScr, A549-shRCC1-A, and A549-shRCC1-B.

### Reduction in RCC1 expression levels across cancer cell lines

Quantitative real-time PCR (qRT-PCR) confirmed significant downregulation of RCC1 mRNA in shRCC1-treated cells. In MDA-MB231 cells, RCC1 expression was reduced by 78% and 79% with shRCC1-A and shRCC1-B, respectively, while in A549 cells, reductions of 65% and 76% were observed for shRCC1-A and shRCC1-B (p-values: 0.0059, 0.0076, 0.0031, and 0.0028) (Fig 1A). This efficient silencing established the reliability of the shRNA system, setting the foundation to explore RCC1's impact on cell viability, proliferation, and migration.

### RCC1 knockdown diminishes cancer cell viability

Using the MTT assay, we assessed the effect of RCC1 silencing on cell viability. Both MDA-MB231 and A549 cells treated with RCC1-targeting shRNAs showed significant reductions in viability at 24, 48, and 72 hours post-transfection (Fig 1B, C). In MDA-MB231 cells, viability dropped by 8% and 13% at 24 hours, progressing to 30% and 25% reductions at 72 hours for shRCC1-A and shRCC1-B, respectively. A549 cells exhibited more pronounced effects, with 39% and 60% viability reductions at 72 hours (p < 0.01). These results indicate that RCC1 supports key survival pathways, and its inhibition significantly compromises cancer cell viability.

### RCC1 silencing impairs long-term proliferation and anchorage-independent growth

To evaluate RCC1's role in sustained cell proliferation, we conducted soft agar colony formation assays, which measure anchorage-independent growth—a hallmark of cancer aggressiveness. RCC1 knockdown significantly inhibited colony formation in both cell lines over seven days (Fig 1D). MDA-MB231-shRCC1-A and shRCC1-B displayed 55% and 50% reductions in colony numbers, while A549-shRCC1-A and shRCC1-B showed 61% and 66% reductions, respectively (p < 0.01). These findings underscore RCC1's critical role in promoting long-term proliferation and tumorigenic potential.

### Impaired migration and invasion following RCC1 silencing

Given RCC1's proposed role in cellular motility, we used Boyden chamber and Matrigel invasion assays to evaluate its impact on migration and invasion. The Boyden chamber assay revealed significant reductions in migration: 45% and 50% for MDA-MB231-shRCC1-A and shRCC1-B, and 50% and 40% for A549-shRCC1-A and shRCC1-B (p = 0.002) (Fig 1E). Matrigel invasion assays corroborated these results, indicating that RCC1 silencing significantly reduced the invasive capacity of cancer cells. These data suggest RCC1 facilitates tumor metastasis by regulating cytoskeletal dynamics and cell adhesion pathways critical for cell motility.

**1A**

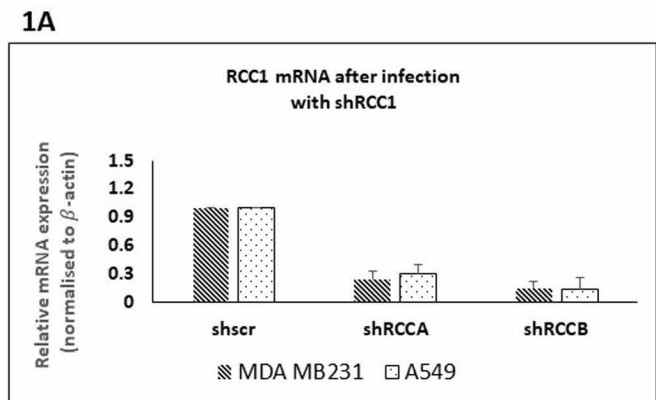

**1B**

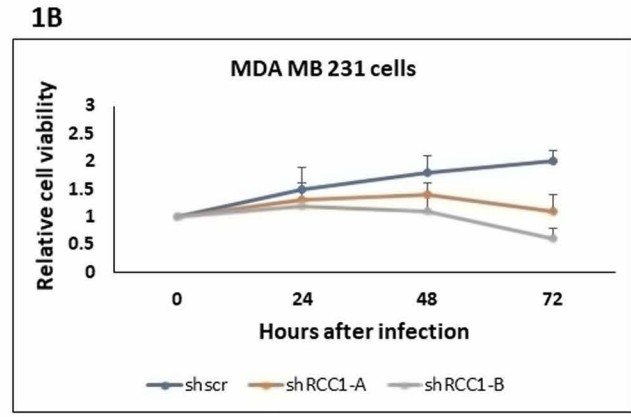

**1C**

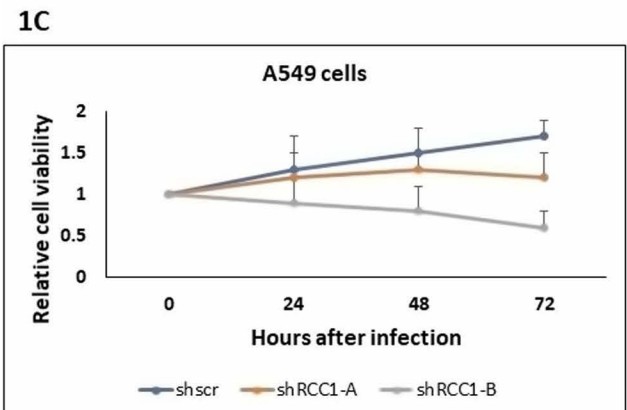
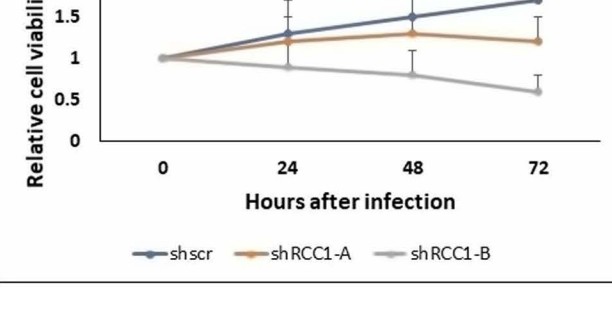

**1D**

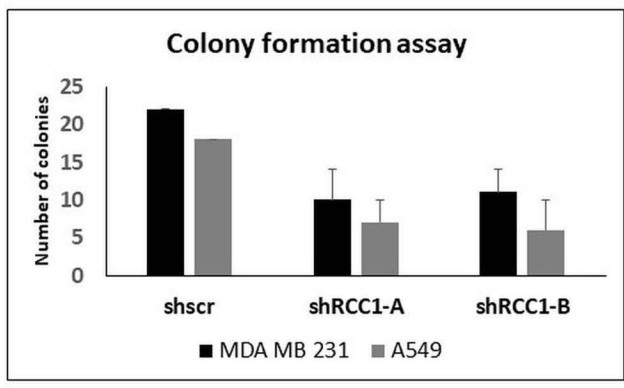

**1E**

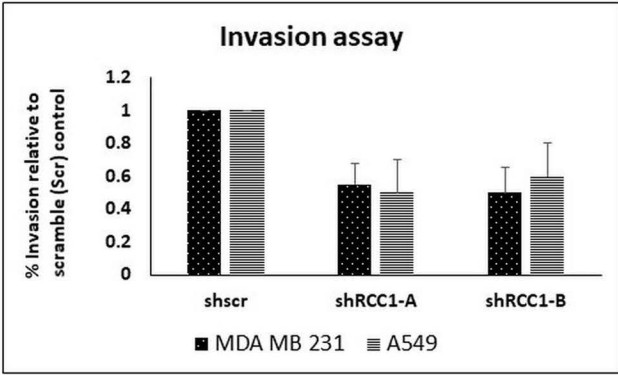

**Fig 1. Knockdown of RCC1 by using shRNA downregulates RCC1 mRNA in MDA-MB-231 and A549 cell lines.** MDA-MB-231 breast cancer and A549 lung cancer cells were analyzed for RCC1 Real-Time PCR (**A**), cell viability (**B & C**), colony formation (**D**), and cell invasion (**E**). Experiments using cells Infected with shScr or shRCC1-A or shRCC1-B.

## Induction of apoptosis and altered expression of apoptotic regulators

Flow cytometric analysis with propidium iodide (PI) staining revealed that RCC1 knockdown increased the proportion of cells in the sub-G1 phase, indicating enhanced apoptosis. Additionally, qRT-PCR showed significant downregulation of the anti-apoptotic gene Bcl2,

with 63% and 61% reductions in shRCC1-A and shRCC1-B cells, respectively (p = 0.003). Conversely, the tumor suppressor gene p53 was upregulated in RCC1-silenced cells, with 44% and 35% increases in MDA-MB231 and A549 cells, respectively (p = 0.003) (Fig 2A-2D). These findings highlight RCC1's role in modulating apoptotic pathways by suppressing pro-apoptotic signals and promoting anti-apoptotic mechanisms.

## RCC1's role in modulating the AKT pathway

To investigate the downstream effects of RCC1 silencing on oncogenic signaling, we analyzed the AKT pathway, a key regulator of cell survival and proliferation. RCC1 knockdown significantly reduced AKT1 and AKT2 mRNA levels in both cell lines. In MDA-MB231-shRCC1-A and shRCC1-B cells, AKT1 levels decreased by 46% and 30%, while AKT2 levels declined by 45% and 40% (Fig 2E). A similar trend was observed in A549 cells. These results suggest that RCC1 modulates the AKT pathway, and its inhibition disrupts critical survival and growth signals, offering therapeutic potential in targeting AKT-driven cancers.

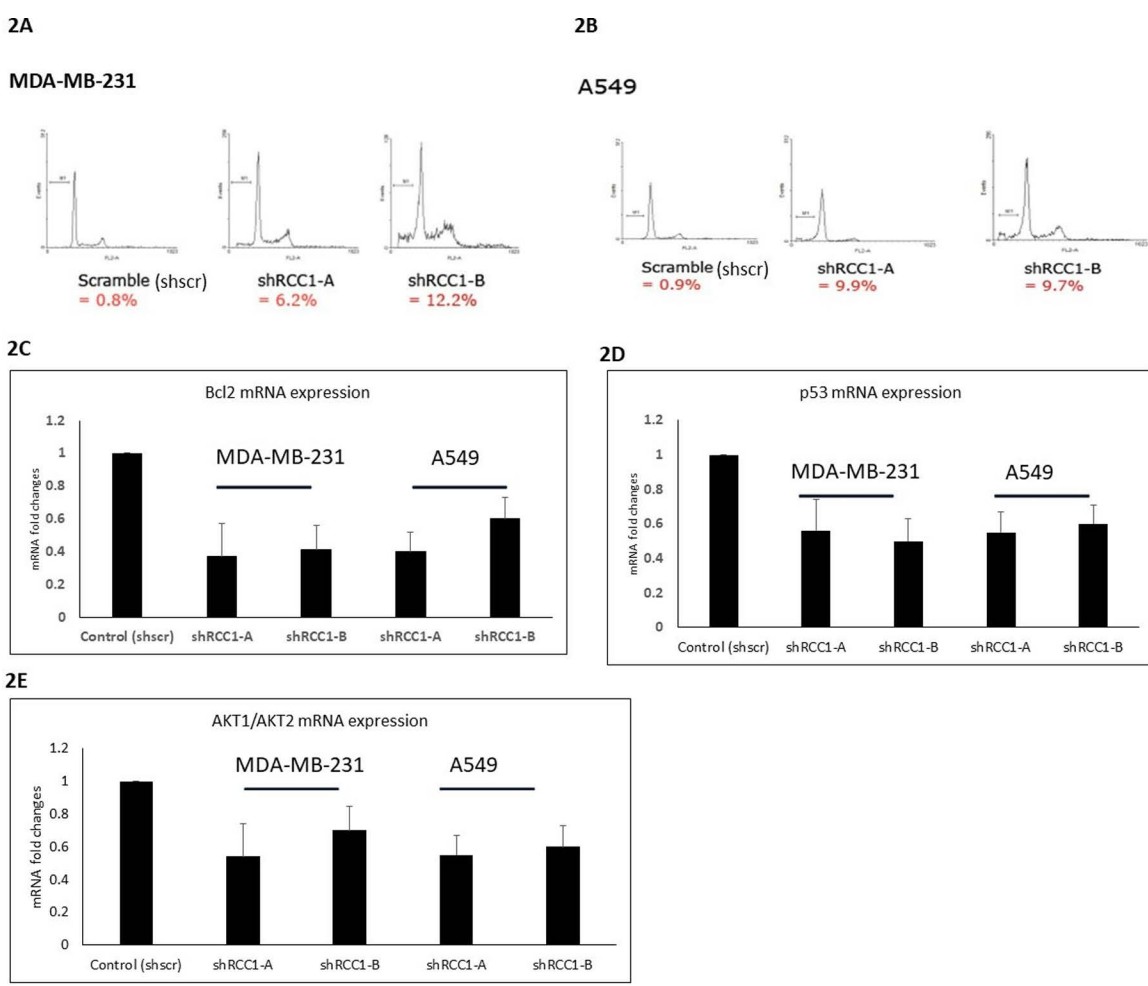

**Fig 2. Knockdown of RCC1 increases cell apoptosis.** For the flow cytometry, **(A)** MDA-MB-231 breast cancer and **(B)** A549 lung cancer cells infected with either shScr or shRCC1, **(C)** lower Bcl2, **(D)** lower p53, and **(E)** lower AKT1 and AKT2. Experiments using cells Infected with shScr or shRCC1-A or shRCC1-B.

## Comprehensive gene expression profiling and pathway dysregulation

Using NimbleGen microarray analysis, we identified 713 differentially expressed genes in RCC1-silenced MDA-MB231 cells, with 368 genes downregulated and 50 upregulated (Fig 3). Hierarchical clustering confirmed distinct expression profiles between RCC1-silenced and control cells. Downregulated genes included SGOL2, involved in chromosomal cohesion, and USP53, associated with stress responses. These dysregulated genes underscore RCC1's integral role in maintaining cancerous traits, such as uncontrolled proliferation and apoptosis evasion.

## Validation of differential gene expression

To confirm microarray results, qRT-PCR was performed on seven downregulated genes, with five showing consistent results (Table 1). Notably, CDC42, a regulator of cell morphology and migration, was significantly downregulated, further linking RCC1 silencing to reduced metastatic potential. This validation supports the robustness of our findings.

## Prognostic significance of RCC1 and associated genes

Kaplan-Meier survival analysis revealed that high RCC1 and SGOL2 expression correlated with poorer survival outcomes in breast and lung cancer patients ($p = 0.00014$ for breast cancer, $p = 3.2e-07$ for lung cancer) (Fig 4A, B, D, E). Conversely, high USP53 expression, inversely regulated by RCC1, was associated with improved survival ($p = 2.9e-0.05$ for breast cancer, $p = 1.3e-11$ for lung cancer) (Fig 4C, F). These findings suggest that RCC1 and SGOL2

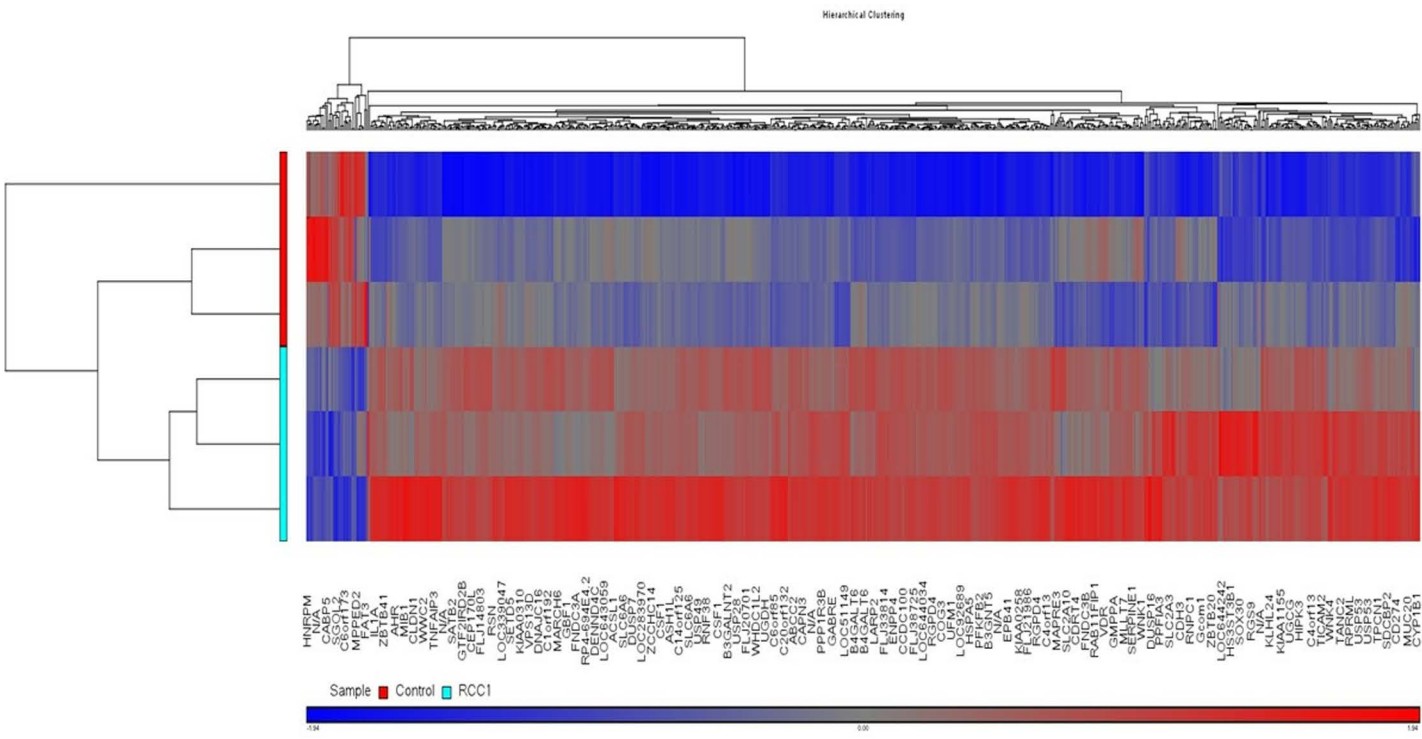

**Fig 3. Knockdown of RCC1 dysregulated RCC1 target genes The study involved three main procedures: (A) RNA microarray analysis, (B) verification of genes that were found to be downregulated, and (C) verification of genes that were found to be elevated.** These procedures were conducted using MDA-MB-231 breast cancer cells that were infected with either shScr or shRCC1.

**Table 1. Correlation analysis of microarray and real-time RT-PCR validation results.**

| Gene Name | Description | Biological Function | Relative fold changes | | | | Microarray | |
|---|---|---|---|---|---|---|---|---|
| | | | MDA MB 231 | P value | MDA MB 231 (RCC1 knock-down) | P value | Fold change (RCC1 vs. Control) | P value |
| Down regulated RCC1 target genes | | | | | | | | |
| CDC42 | Cell division cycle 42, small GTPase of the Rho-subfamily | Regulates signaling pathways that control diverse cellular functions including cell morphology, migration, endocytosis and cell cycle progression | 2.51219 | 0.0206 | 0.66617 | 0.00345 | -2.07873 | 4.34E-05 |
| TUBA3 | Tubulin alpha 1a (belong to the tubulin superfamily) | Nucleation of microtubule assembly. | 3.09032 | 0.8499 | 4.25281 | 0.40116 | -2.72153 | 0.000607096 |
| HNRPM | Heterogeneous nuclear ribonucleoprotein M | Controls circRNA biogenesis and splicing fidelity to sustain cancer cell fitness | 0.79663 | 0.0210 | 0.00359 | 0.00070 | -2.10446 | 0.000689007 |
| SIVA1 | Apoptosis inducing factor | Mediate the ubiquitination of proliferating cell nuclear antigen (PCNA), an important step in translesion DNA synthesis. | 0.68473 | 0.9801 | 0.63560 | 0.83742 | -2.20867 | 0.00252951 |
| HIST1H1E | H1.4 linker histone, cluster member | Interacts with linker DNA between nucleosomes and functions in the compaction of chromatin into higher order structures | 3.24848 | 0.1927 | Undetectable | ---- | -2.45382 | 0.0029549 |
| SGOL2 | Shugoshin 2 | Involved in homologous chromosome segregation; meiotic sister chromatid cohesion; and mitotic sister chromatid segregation | 5.08311 | 0.8506 | 0.03804 | 0.06761 | -2.35731 | 0.00379092 |
| NAP1L3 | Nucleosome assembly protein 1 like 3 | Regulating chromatin structure and consequently transcriptional gene regulation | 1.9276 | 0.9249 | 0.00076 | 0.07353 | -2.03287 | 0.00333086 |
| Up regulated RCC1 target genes | | | | | | | | |
| FBXW10 | F-box and WD repeat domain containing 10 | Promotes hepatocarcinogenesis in male patients and mice | 5.8464 | 0.0007 | 2.38 | 0.2260 | 4.4176 | 0.000384187 |
| USP53 | Ubiquitin specific peptidase 53 | Inhibits the occurrence and development of clear cell renal cell carcinoma through NF-kappaB pathway inactivation | 7.0070 | <0.0001 | 9.0915 | 0.7604 | 3.33442 | 0.000563702 |
| KIAA1024 | Membrane integral NOTCH2 associated receptor 1 | Involved in several processes, including negative regulation of TOR signaling; negative regulation of angiogenesis; and negative regulation of protein ubiquitination. Located in plasma membrane | 20.306 | <0.0001 | 4.799 | <0.0001 | 3.00228 | 0.000898928 |
| SOD2 | Superoxide dismutase 2 (iron/manganese superoxide dismutase family) | Binds to the superoxide byproducts of oxidative phosphorylation and converts them to hydrogen peroxide and diatomic oxygen | 15.003 | <0.0001 | 7.8515 | 0.0020 | 3.60097 | 0.00140377 |
| GSDML | Gasdermin B | Implicated in the regulation of apoptosis in epithelial cells, and are linked to cancer | 6.38337 | 0.0003 | 1.554 | 0.0413 | 3.52921 | 0.00175397 |
| MAPRE3 | Microtubule associated protein RP/EB family member 3 | Involved in positive regulation of cyclin-dependent protein serine/threonine kinase activity and positive regulation of transcription, DNA-templated | Undetectable | --- | 4.299 | ---- | 3.04144 | 0.00104763 |
| ABCC2 | ATP binding cassette subfamily C member 2 | Appears to contribute to drug resistance in mammalian cell | 0.34851 | 0.9923 | 1.4615 | 0.1272 | 3.15057 | 0.00169749 |
| SOX30 | SRY-box transcription factor 30 | Regulation of embryonic development and in the determination of the cell fate | 3.0626 | 0.3706 | 2.38 | 0.9192 | 3.53071 | 0.00248036 |

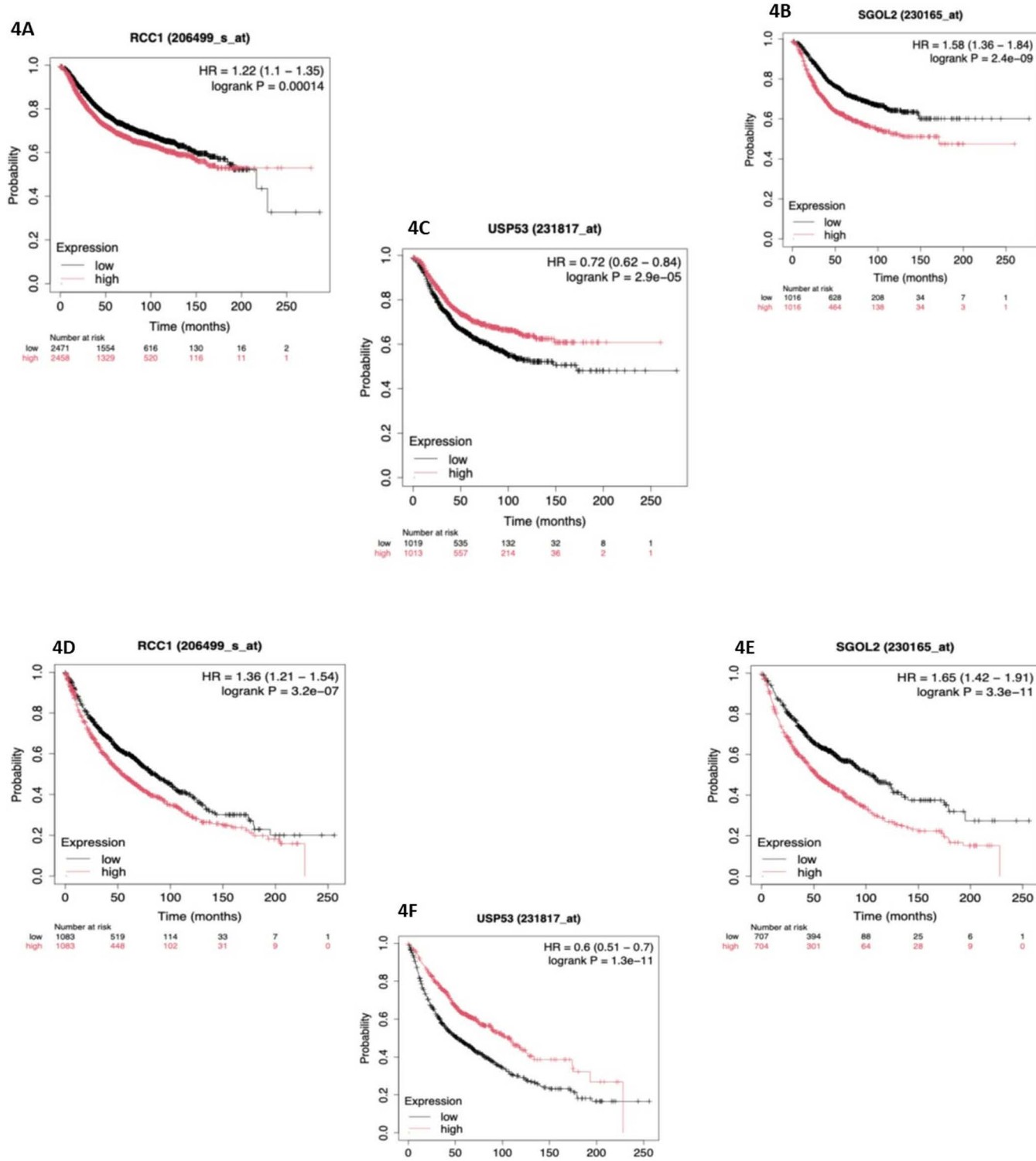

**Fig 4. Kaplan–Meier long-term survival curves (** https://kmplot.com/analysis/index.php?p=service**) grouped by high and low RCC1, SGOL2 and USP53 expressions in breast and lung patients. (A, B & C)** probability in breast cancer patients and **(D, E & F)** probability in lung cancer patient's comparison analysis of patients with high and low RCC1, SGOL2 and USP53 expressions.

may serve as markers of aggressive cancer phenotypes, while USP53 indicates a more favorable prognosis.

### Differential expression in tumor versus normal tissues

Gene Expression Profiling Interactive Analysis (GEPIA) datasets confirmed RCC1 and SGOL2 overexpression in tumors compared to normal tissues, while USP53 was consistently downregulated (Fig 5A, B, C). These patterns reinforce RCC1's classification as an oncogene and USP53's potential role as a tumor suppressor, further validating RCC1's therapeutic relevance.

## Discussion

A growing body of evidence emphasizes the critical role of RCC1 in tumor biology, primarily through its regulation of the cell cycle and influence on tumor progression [16,30–33]. In our study, RCC1 was found to play a multifaceted role in the development of breast and lung cancers by promoting cell viability, migration, and invasion while inhibiting apoptotic responses. Knockdown of RCC1 in MDA-MB-231 and A549 cell lines resulted in substantial reductions in cell survival, proliferation, and metastatic potential, demonstrated by decreased viability, colony formation, migration, and invasion. These findings underscore RCC1's pivotal contribution to oncogenic behaviors, likely mediated through pathways involving cell cycle regulators and apoptosis suppression.

RCC1's role in cell cycle regulation is well established, functioning as a guanine nucleotide exchange factor (GEF) for Ran, a small GTPase involved in processes such as chromatin condensation, nuclear envelope reassembly, and spindle formation [34,35]. The reduced viability and colony formation observed upon RCC1 knockdown suggest its role in supporting

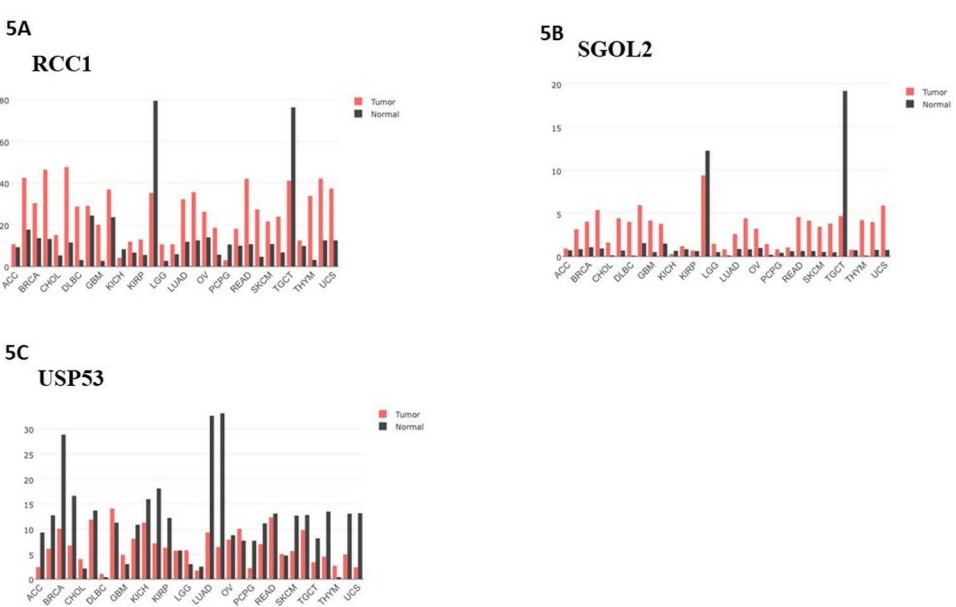

**Fig 5. Expression of RCC1, SGOL2 and USP53 in various human tumours.** The Gene_DE module revealed differential (**A**) RCC1, (**B**) SGOL2 and (**C**) USP53 expressions between tumour tissues and adjacent normal tissues across all GEPIA tumours (http://gepia.cancer-pku.cn/detail.php?gene). Plots were used to display gene expression level distribution.

unregulated cell cycle progression in cancer cells. This aligns with prior studies showing that RCC1 expression promotes the dysregulated G1/S cell cycle transition commonly observed in cancer [15,21].

Our findings also highlight the significant impact of RCC1 on metastatic behavior, as RCC1-silenced cells exhibited diminished migration and invasion capabilities in both breast and lung cancer models. Metastasis, the leading cause of cancer-related mortality, involves intricate processes, including cell adhesion, motility, and interaction with the extracellular matrix (ECM). RCC1 appears to facilitate these processes by enhancing cancer cell dissemination and tissue infiltration. The observed reduction in migration and invasion upon RCC1 knockdown suggests that RCC1 regulates actin cytoskeleton remodeling and adhesion molecule expression, crucial for cell motility [25,27]. Additionally, RCC1's influence on the phosphorylation of motility proteins, such as Rac1 and Cdc42, further explains its regulatory role in cell migration and invasion [36]. These findings implicate RCC1 as a key regulator of cellular machinery that supports cancer metastasis.

Importantly, RCC1 knockdown significantly altered apoptotic signaling pathways, evidenced by reduced expression of the anti-apoptotic protein Bcl2 and increased expression of the tumor suppressor protein p53. Bcl2 prevents mitochondrial membrane permeabilization, a critical step in apoptosis initiation, and its downregulation following RCC1 knockdown underscores RCC1's role in modulating anti-apoptotic mechanisms to promote cancer cell survival [22,23]. Conversely, the observed upregulation of p53 following RCC1 suppression aligns with increased apoptosis, highlighting RCC1's interference with tumor suppressor pathways. Our data also revealed an increase in sub-G1 phase cells after RCC1 knockdown, indicating heightened apoptosis. Apoptosis, a programmed cell death mechanism essential for maintaining tissue homeostasis, is often evaded by cancer cells to support tumor growth and therapeutic resistance. RCC1 appears to facilitate this evasion by suppressing pro-apoptotic pathways while enhancing anti-apoptotic signals. These observations align with previous studies reporting RCC1's role in promoting resistance to pro-apoptotic stimuli, including those induced by therapeutic agents [37, 38]. The increased expression of p53 suggests that RCC1 knockdown induces cellular stress responses, sensitizing cancer cells to intrinsic apoptotic pathways. The balance between Bcl2 and p53 is a critical determinant of cell fate, and RCC1's modulation of this axis may explain its role in promoting cancer cell survival under stress conditions [39].

Given its role in regulating the cell cycle, apoptosis resistance, and metastatic behavior, RCC1 represents a potential target for further investigation in cancer therapy. Knockdown experiments demonstrated that RCC1 suppression impairs cancer cell survival and invasiveness in vitro, suggesting that targeting RCC1 could provide insights into its role in oncogenic pathways. Further studies are needed to determine whether RCC1 inhibition, in combination with other treatments such as chemotherapeutics or targeted agents, could influence cancer cell proliferation and metastasis. Emerging research suggests that RCC1 may also influence immune checkpoint proteins like PD-L1, potentially modulating immune evasion [22]. This opens the possibility of combining RCC1 inhibitors with immunotherapies, particularly in lung cancer, where immune checkpoint inhibitors have shown clinical success [40].

RCC1's overexpression in several cancer types and its association with aggressive phenotypes suggest its potential as a prognostic biomarker, though further validation is required. Studies have shown that high RCC1 expression correlates with poor survival outcomes in cancers such as breast and lung cancer, likely due to its role in enhancing proliferation, invasion, and apoptosis resistance [15,20]. Our findings support this, as RCC1 knockdown significantly reduced metastatic and proliferative capacities, implicating it in pathways associated with adverse clinical outcomes. RCC1, along with genes like SGOL2 and USP53, could form a

prognostic biomarker panel to assess cancer aggressiveness and predict patient outcomes. Such a panel would allow clinicians to tailor treatment strategies based on RCC1 expression levels, enabling more personalized approaches to cancer care. Future research should focus on developing RCC1-based biomarker assays to improve diagnostic accuracy and guide therapeutic decision-making.

Our study demonstrates that RCC1 is a crucial regulator in breast and lung cancer progression, influencing key cellular processes such as survival, proliferation, and metastasis. The significant reduction in cell viability, migration, and invasion upon RCC1 knockdown underscores its essential role in promoting oncogenic characteristics. By acting as a GEF for Ran, RCC1 facilitates cell cycle progression and apoptosis resistance, enabling cancer cells to thrive under stress conditions. Additionally, RCC1's role in modulating apoptotic pathways, as evidenced by the downregulation of Bcl2 and upregulation of p53, shifts the cellular balance toward cell death upon RCC1 knockdown. This highlights RCC1's critical function in evading programmed cell death, a hallmark of cancer resilience. Furthermore, the observed decrease in metastatic behavior in RCC1-silenced cells suggests its involvement in cell adhesion and motility, essential for cancer dissemination.

Given its multifaceted roles, RCC1 warrants further investigation as a potential therapeutic target in cancer research. Combining RCC1-targeted therapies with existing treatment regimens could enhance therapeutic efficacy and overcome drug resistance in aggressive cancers. Additionally, RCC1's association with poor prognostic outcomes suggests its potential utility in patient stratification, though additional studies are needed to confirm its role in personalized treatment approaches. This study not only elucidates the critical functions of RCC1 in cancer biology but also provides a foundation for future research into its role across various cancer types and interactions with therapeutic agents [41–43].

One potential limitation of targeting RCC1 is the risk of off-target effects due to its essential role in cellular processes. As RCC1 is highly conserved, its inhibition may impact normal cell function, particularly in rapidly dividing cells. To mitigate these effects, the development of selective targeting strategies, such as tumor-specific delivery systems and highly specific RCC1 inhibitors, is essential. Our findings provide substantial evidence of RCC1's involvement in key aspects of cancer cell biology. By promoting survival, proliferation, and metastatic behavior, RCC1 acts as a central regulator in cancer progression. Targeting RCC1 in cancer models may help elucidate its role in oncogenic pathways, offering insights into potential therapeutic strategies for aggressive and treatment-resistant cancers. While RCC1 represents a promising therapeutic target for disrupting multiple oncogenic pathways, its inhibition may pose potential risks due to its fundamental role in essential cellular processes. RCC1 functions as a key regulator of the Ran GTPase cycle, influencing nucleocytoplasmic transport, mitotic spindle assembly, and chromatin dynamics processes that are not exclusive to cancer cells but are also crucial for normal cellular homeostasis. Consequently, systemic RCC1 inhibition could lead to unintended cytotoxic effects in rapidly proliferating normal tissues, particularly those with high mitotic activity, such as bone marrow, gastrointestinal epithelium, and skin. Additionally, RCC1 suppression may disrupt nuclear transport and mitotic checkpoint fidelity, potentially leading to genomic instability and adverse effects on non-cancerous cells. Furthermore, given RCC1's interactions with signaling pathways like AKT and Bcl2, its inhibition could inadvertently trigger compensatory survival mechanisms that promote therapy resistance or alter apoptotic thresholds in non-cancerous tissues. Another critical gap in current knowledge is the role of RCC1 in non-cancerous models, which remains largely unexplored. Future studies should investigate the broader physiological impact of RCC1 inhibition, identify potential toxicities in normal tissues, and develop tumor-specific targeting strategies such as nanoparticle-based delivery

or selective small-molecule inhibitors to minimize off-target effects. Addressing these challenges will be crucial to advancing RCC1-based therapies while ensuring their safety and clinical viability.

RCC1's association with aggressive phenotypes further underscores its potential as a prognostic biomarker. High RCC1 expression may indicate poor prognosis, allowing for stratification of patients based on RCC1 levels and personalized treatment planning. Incorporating RCC1 into biomarker panels, alongside SGOL2 and USP53, could improve diagnostic accuracy and enable more effective, targeted cancer treatments. Future studies should explore RCC1's impact on other cancer types, evaluate its interactions with therapeutic agents, and investigate combination strategies to enhance efficacy, particularly in immune-sensitive cancers. By targeting RCC1-dependent cancer survival mechanisms, combination therapies with RCC1 inhibitors and standard chemotherapeutics could provide robust treatment options, enhance therapeutic outcomes, and address chemoresistance challenges.

In conclusion, our study highlights the potential role of RCC1 in cancer biology, particularly in breast and lung cancer models. RCC1 inhibition decreased cancer cell viability, invasion, and colony formation while inducing apoptosis, making it a promising candidate for targeted therapies. When combined with existing treatment regimens, RCC1 inhibitors could enhance efficacy, especially in cancers with high RCC1 expression. Overall, RCC1's multifunctional role in cancer survival and progression suggests it may be a potential biomarker and therapeutic target, though further preclinical and clinical research is needed.

## Supporting information

**S1 Table. Differential Gene Expression in RCC1-Silenced MDA-MB-231 Cells.** The table lists 713 genes with ≥ 2-fold changes in expression in MDA-MB-231-shRCC1 cells compared to controls, including probe set ID, gene name, p-value, and fold-change. Statistical significance (p-values) and expression magnitude (fold-change) are detailed.
(XLSX)

## Acknowledgement

Authors are grateful to the RAK Medical and Health Sciences University for providing the state-of-the-art research facility and seed fund to carry out this research project.

## Author contributions

**Conceptualization:** Mohamed El-Tanani.

**Data curation:** Mohamed El-Tanani, Shakta Mani Satyam, Razan M. Obeidat, Yahia El-Tanani, Taher Hatahet.

**Formal analysis:** Mohamed El-Tanani, Shakta Mani Satyam, Syed Arman Rabbani, Razan M. Obeidat, Yahia El-Tanani, Alaa A. A. Aljabali, Taher Hatahet.

**Funding acquisition:** Mohamed El-Tanani.

**Investigation:** Mohamed El-Tanani, Shakta Mani Satyam, Razan M. Obeidat.

**Methodology:** Mohamed El-Tanani, Shakta Mani Satyam, Syed Arman Rabbani, Razan M. Obeidat, Yahia El-Tanani, Alaa A. A. Aljabali, Taher Hatahet.

**Project administration:** Mohamed El-Tanani, Shakta Mani Satyam, Razan M. Obeidat, Yahia El-Tanani, Alaa A. A. Aljabali, Taher Hatahet.

**Resources:** Mohamed El-Tanani, Shakta Mani Satyam, Razan M. Obeidat, Yahia El-Tanani, Alaa A. A. Aljabali.

**Software:** Mohamed El-Tanani, Shakta Mani Satyam, Syed Arman Rabbani, Razan M. Obeidat, Yahia El-Tanani, Alaa A. A. Aljabali, Taher Hatahet.

**Supervision:** Mohamed El-Tanani, Shakta Mani Satyam.

**Validation:** Mohamed El-Tanani, Shakta Mani Satyam, Syed Arman Rabbani, Razan M. Obeidat, Yahia El-Tanani, Alaa A. A. Aljabali, Taher Hatahet.

**Visualization:** Mohamed El-Tanani, Shakta Mani Satyam, Syed Arman Rabbani, Razan M. Obeidat, Yahia El-Tanani, Alaa A. A. Aljabali, Taher Hatahet.

**Writing – original draft:** Mohamed El-Tanani, Shakta Mani Satyam.

**Writing – review & editing:** Mohamed El-Tanani, Shakta Mani Satyam, Syed Arman Rabbani, Razan M. Obeidat, Yahia El-Tanani, Alaa A. A. Aljabali, Taher Hatahet.

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
