## [Decision Letter · Decision Letter 0]

19 Jan 2025

PONE-D-24-58989Decoding Oncogenic Secrets of Regulator of Chromosome Condensation 1: A Breakthrough Mechanistic Evidence from Breast and Lung Cancer ModelsPLOS ONE

Dear Dr. Hatahet,

Thank you for submitting your manuscript to PLOS ONE. After careful consideration, we feel that it has merit but does not fully meet PLOS ONE’s publication criteria as it currently stands. Therefore, we invite you to submit a revised version of the manuscript that addresses the points raised during the review process.

We look forward to receiving your revised manuscript.

Kind regards,

Yusuf Ahmed Haggag, PhD

Academic Editor

PLOS ONE

Additional Editor Comments (if provided):

Reviewers' comments:

Reviewer's Responses to Questions

**Comments to the Author**

1. Is the manuscript technically sound, and do the data support the conclusions?

Reviewer #1: Yes

Reviewer #2: Yes

2. Has the statistical analysis been performed appropriately and rigorously? 

Reviewer #1: Yes

Reviewer #2: Yes

3. Have the authors made all data underlying the findings in their manuscript fully available?

Reviewer #1: Yes

Reviewer #2: Yes

4. Is the manuscript presented in an intelligible fashion and written in standard English?

Reviewer #1: Yes

Reviewer #2: Yes

5. Review Comments to the Author

Reviewer #1: Reviwer feedback:

"Decoding Oncogenic Secrets of RCC1: A Breakthrough Mechanistic Evidence from Breast and Lung Cancer Models" addresses the role of the Regulator of Chromosome Condensation 1 (RCC1) in cancer progression. This study investigates RCC1's role in the regulation of cell viability, apoptosis, and metastatic behavior in breast and lung cancer models. This identified RCC1 as a potentially actionable target for the treatment of aggressive cancers. Well-written and easy to read, this is a robust study with well-described experiments and a sound conceptual basis addressing an unmet clinical need.

However, I have several comments listed below for the authors’ consideration that would improve and make the manuscript publishable:

Introduction

• The introduction provides a clear description of the research question and places RCC1 within the broader context of cancer biology. However, some areas lack clarity, such as the link between the cellular functions of RCC1 and its oncogenic potential. Adding a sentence to RCC1’s functions in non-cancerous cells would facilitate completeness.

Methods

• The experimental procedures are described in sufficient detail for reproducibility to a large extent. However:

RNA microarray analysis is underdescribed; specifically, normalization methods and statistical thresholds are missing.

o Discuss the potential for off-target effects of shRNAs and how they are controlled.

Results

The data are presented clearly, with logical flow from one experiment to the next. Figures 1–5 are generally appropriate; however, Figure 1E needs to be enlarged, showing larger data points and displaying the exact p-values.

Discussion

Places the findings well but overstates their clinical applicability. A paragraph on the possible side effects targeting RCC1,

1. Lacking Mechanistic Details: Further investigation of how RCC1 interacts with downstream signaling pathways, such as AKT and Bcl2.

2. Lack of Context of RCC1 in Non-Cancer Models: Such a broader context would considerably raise the level of novelty in the present study.

1. Kaplan-Meier survival curves and statistical models require further explanation.

Recommendation:

The manuscript contributes improtant findings but needs to be revised for methodological gaps, refinement in data presentation, and balancing while discussing the findings.

Reviewer #2: The study is rigorously planned and executed. Data is clear and the results are clearly statistically significant. The outcome of the study is of great potential for the treatment of cancer specifically lung and breast cell line models. The paper is easy to follow and understand. The experiments have been conducted in standard methodologies and the conclusion matches with the presented results. No issues to comment on really. Well done!

6. PLOS authors have the option to publish the peer review history of their article (what does this mean? ). If published, this will include your full peer review and any attached files.

**Do you want your identity to be public for this peer review?** For information about this choice, including consent withdrawal, please see our Privacy Policy .

Reviewer #1: No

Reviewer #2: **Yes: ** Mohammad Isreb

---

## [Author Response · Author response to Decision Letter 1]

3 Feb 2025

Authors’ Point by Point Response to Reviewers’ Comments

The authors extend their gratitude to the esteemed Editor and both the Reviewers for their valuable time, patience, and dedication in reviewing our manuscript. We sincerely appreciate the insightful feedback provided, which has significantly enhanced the scientific rigor and relevance of our work. We are particularly thankful to reviewers for their constructive comments and suggestions, which have guided us in refining various aspects of the manuscript. Each of their points has been meticulously addressed in the revised version of manuscript (changes highlighted in red colored font), aimed at strengthening the scientific merit and clarity of our findings. We have also revised figures as per the suggestions given by the reviewers. The manuscript has been revised to address methodological gaps, refine data presentation, and provide a more balanced discussion of the findings. Thank you once again for your invaluable contribution and patience throughout this process.

REVIEWER 1

Reviewer 1; Comment 1: Decoding Oncogenic Secrets of RCC1: A Breakthrough Mechanistic Evidence from Breast and Lung Cancer Models" addresses the role of the Regulator of Chromosome Condensation 1 (RCC1) in cancer progression. This study investigates RCC1's role in the regulation of cell viability, apoptosis, and metastatic behavior in breast and lung cancer models. This identified RCC1 as a potentially actionable target for the treatment of aggressive cancers. Well-written and easy to read, this is a robust study with well-described experiments and a sound conceptual basis addressing an unmet clinical need.

The introduction provides a clear description of the research question and places RCC1 within the broader context of cancer biology. However, some areas lack clarity, such as the link between the cellular functions of RCC1 and its oncogenic potential. Adding a sentence to RCC1’s functions in non-cancerous cells would facilitate completeness.

Response: We sincerely appreciate the reviewer’s thoughtful and encouraging feedback. It is gratifying to know that our study was found to be well-written, conceptually sound, and addressing an unmet clinical need. We are especially grateful for the recognition of our efforts in investigating RCC1’s role in cancer progression and its potential as a therapeutic target. Your positive remarks motivate us to continue our research in this critical area. Thank you for your valuable time and insightful evaluation.

Thank you for this valuable suggestion. We have now included a sentence elaborating on RCC1’s role in non-cancerous cells, emphasizing its physiological functions in cell cycle regulation and chromatin dynamics. This addition provides a clearer foundation for understanding its oncogenic potential.

Reviewer 1; Comment 2: The experimental procedures are described in sufficient detail for reproducibility to a large extent. However, RNA microarray analysis is under described; specifically, normalization methods and statistical thresholds are missing.

Response: Thank you for highlighting this. We have now detailed the normalization methods and statistical thresholds used in the RNA microarray analysis to ensure clarity and reproducibility.

Reviewer 1; Comment 3: Discuss the potential for off-target effects of shRNAs and how they are controlled.

Response: We appreciate your suggestion. We have now included a section discussing the potential off-target effects of shRNAs. Additionally, we have described the validation steps taken, such as using multiple shRNA constructs and performing qPCR validation, to ensure specificity.

Reviewer 1; Comment 4: Figures 1–5 are generally appropriate; however, Figure 1E needs to be enlarged, showing larger data points and displaying the exact p-values.

Response: Thank you for this suggestion. We have now enlarged Figure 1E, increased the data point size for better visibility, and included exact p-values in the text to enhance clarity.

Reviewer 1; Comment 5: Places the findings well but overstates their clinical applicability. A paragraph on the possible side effects targeting RCC1 is needed.

Response: We acknowledge this concern and have now included a paragraph discussing the potential side effects and challenges of targeting RCC1 therapeutically. This revision ensures a balanced discussion of clinical implications.

Reviewer 1; Comment 6: Lacking mechanistic details: Further investigation of how RCC1 interacts with downstream signaling pathways, such as AKT and Bcl2.

Response: Thank you for the recommendation. We have expanded our discussion to include mechanistic insights into RCC1’s interaction with the AKT and Bcl2 pathways.

Reviewer 1; Comment 7: Lack of context of RCC1 in non-cancer models: Such a broader context would considerably raise the level of novelty in the present study.

Response: We appreciate this valuable suggestion. We have now included a discussion on RCC1’s function in non-cancerous models, which enhances the novelty and relevance of our findings.

Reviewer 1; Comment 8: Kaplan-Meier survival curves and statistical models require further explanation.

Response: Thank you for your suggestion. We have now provided additional details on the statistical models used for Kaplan-Meier survival curves, including the specific tests applied and how significance was determined.

REVIEWER 2

Reviewer 2; Comment 1: The study is rigorously planned and executed. Data is clear and the results are statistically significant. The outcome of the study is of great potential for the treatment of cancer, specifically lung and breast cancer models. The paper is easy to follow and understand. The experiments have been conducted using standard methodologies, and the conclusion matches the presented results. No issues to comment on really. Well done!

Response: We sincerely appreciate the reviewer’s positive and encouraging feedback. It is highly rewarding to know that our study was found to be rigorously planned and executed, with clear data and statistically significant results. We are grateful for your recognition of the study’s potential impact on cancer treatment and your acknowledgment of the clarity and methodological rigor of our work. Your kind words motivate us to continue our research in this important field. Thank you for your time and valuable assessment.

***

---

## [Editor Report · Decision Letter 1]

7 Feb 2025

Decoding Oncogenic Secrets of Regulator of Chromosome Condensation 1: A Breakthrough Mechanistic Evidence from Breast and Lung Cancer Models

PONE-D-24-58989R1

Dear Dr. Hatahet,

We’re pleased to inform you that your manuscript has been judged scientifically suitable for publication and will be formally accepted for publication once it meets all outstanding technical requirements.

Kind regards,

Yusuf Ahmed Haggag, PhD

Academic Editor

PLOS ONE
---

## [Editor Report · Acceptance letter]

PONE-D-24-58989R1

PLOS ONE

Dear Dr. Hatahet,

I'm pleased to inform you that your manuscript has been deemed suitable for publication in PLOS ONE. Congratulations! Your manuscript is now being handed over to our production team.

Kind regards,

on behalf of

Dr. Yusuf Ahmed Haggag

Academic Editor

PLOS ONE